# COVID-19 Vaccine for Children: Determinants and Beliefs Contributing to Vaccination Decision of Parents in Germany 2021/2022

**DOI:** 10.3390/vaccines12010020

**Published:** 2023-12-23

**Authors:** Laura Purrmann, Leoni-Johanna Speichert, Alexander Bäuerle, Martin Teufel, Julia Barbara Krakowczyk, Jil Beckord, Ursula Felderhoff-Müser, Eva-Maria Skoda, Hannah Dinse

**Affiliations:** 1Clinic for Psychosomatic Medicine and Psychotherapy, LVR-University Hospital Essen, University of Duisburg-Essen, 45147 Essen, Germany; 2Centre for Translational Neuro- and Behavioral Sciences (C-TNBS), University of Duisburg-Essen, 45147 Essen, Germany; 3Department of Pediatrics I, University Hospital of Essen, University of Duisburg-Essen, 45147 Essen, Germany

**Keywords:** COVID-19, willingness, trust, childhood vaccines, associated factors, vaccination, mental health

## Abstract

To reduce the number of COVID-19 cases, vaccines were rapidly made available worldwide. For a strategically targeted response to the COVID-19 pandemic, population vaccination coverage was to be maximized. The target groups also included healthy children. In this context, it is important to understand the determinants and beliefs that lead parents to favor or oppose COVID-19 immunization in children. This study aimed to investigate parents’ COVID-19 vaccination willingness in Germany for children aged 5–11 years in 2021/2022. For this purpose, the determinants and beliefs behind parents’ vaccination decisions were examined. Descriptive analysis and bivariate correlations were performed on COVID-19 vaccination willingness and parents’ mental health status, general vaccination attitudes, and SARS-CoV-2 politics perceptions. In total, 2401 participants fully participated in this cross-sectional study. The COVID-19 vaccination uptake (71.4%) outweighed the vaccination refusal (19.4%). Correlations revealed higher vaccine acceptance in parents presenting full vaccination certificates (90.9%), COVID-19 immunizations (99.9%), or increased COVID-19 fear (93.6%). Vaccination-refusal was associated with higher perceived pressure by COVID-19 vaccination campaigns (87.7%), higher experienced restrictions due to COVID-19 protective measures in parents’ social environment (83.6%), and engagement against COVID-19 protective measures (51.6%). Besides general anxiety, no significant correlations were observed between parents’ mental health variables and vaccination willingness. Although several factors are ultimately associated with vaccination willingness, future vaccination campaigns should prioritize reducing pressure, increasing trust, and considering parents’ differentiation between familiar and unfamiliar pathogens during their vaccination decision-making process.

## 1. Introduction

Over the past three years, society has learned to cope with the SARS-CoV-2 virus and the presence of the COVID-19 pandemic has gradually faded into the background. To date, there is no exclusive target therapy to effectively combat the virus and in addition to protective COVID-19 vaccinations, potential drugs are still being improved to adequately treat severe courses of the disease [1,2]. COVID-19 may present various clinical manifestations and undergo different courses [3]. Current research focuses on conditions that can develop after COVID-19 infections, for example, post COVID-19 syndrome, and investigates the association between such conditions and the presence of COVID-19 vaccinations [4,5,6]. Due to initially insufficient effective and available therapies, parallel increasing death rates, and virus variants, a worldwide rollout of vaccines was the only solution to achieve a sufficient, population-wide immunity and protection against SARS-CoV-2 in the short term through prevention [7].

Based on World Health Organization (WHO) evaluations, vaccination campaigns have been rapidly established worldwide to achieve herd immunity in the population. In Germany, the first vaccination invitations for high-risk groups (elderly, pre-diseased adults, health workers) were issued in December 2020 [8,9]. Later vaccination recommendations were successively extended for healthy adults and by August 2021 for teenagers. By November 2021, the European Medical Agency approved the vaccine BioNTech/Pfizer (Comirnaty^®^ 10 μg) for 5–11-year-olds in Europe [8,10,11,12]. This was followed by a vaccination recommendation in Germany issued by the Standing Committee on Vaccination (STIKO) in May 2022. However, this recommendation was withdrawn after one year for all (non-pre-diseased) minors [8,13,14]. This change was justified based on updated WHO vaccination declarations. According to WHO, SARS-CoV-2 causes less severe courses in healthy children and their hospitalization and death rates are lowest, requiring high numbers needed to vaccinate [14,15]. Furthermore, it is not necessarily cost effective for countries to vaccinate healthy children [14,15]. Despite milder courses or inapparent infections among children, it should not be neglected that severe courses may occur. Children may transmit the infection to their (vulnerable) environment and may suffer long-term consequences [15]. Current components of COVID-19 studies in children are pediatric inflammatory multisystem syndrome, long COVID-19, and post COVID-19 syndromes [8,16,17]. Although these consequences are less frequent in children, COVID-19 vaccination uptake for children remains a controversial issue for both parents and authorities [8]. The monitoring of German COVID-19 vaccination rates reported that 22.4% of the 5.4 million 5–11-year-olds in Germany had received at least one vaccination shot by November 2023 [18,19]. The importance of vaccinations in general becomes more serious considering the observed decline in routine childhood vaccines and the resulting worrying vaccination gaps during 2019–2021 [20,21,22,23]. According to UNICEF, “67 million children missed out on their routine vaccinations” during the pandemic [20]. In parallel, measles infection rates rose last year, and health officials expect devastating disease outbreaks in the upcoming years due to vaccination backlogs [20,24].

Considering these developments, it is important to understand why vaccination gaps for routine childhood vaccinations have occurred and why demand has been low, particularly concerning COVID-19 vaccine uptake among children during the pandemic. To provide an attractive vaccination offer to parents and to provide education, parents’ motivations for their vaccination decisions should be elicited. To this end, it is necessary to identify the most important factors involved in parents’ considerations for or against vaccination uptake for children. International studies therefore examined vaccination acceptance and hesitancy in the context of COVID-19 vaccines and concluded the general attitudes toward vaccination, the COVID-19 immunization status among parents themselves, and the sociodemographic variables that strongly influence vaccination decisions for children [25,26,27]. Other factors may relate to radical protective measures (e.g., lockdowns), fast-moving as well as far-reaching policy decisions, political polarization, and misinformation about vaccines during the COVID-19 pandemic [23,25,28,29,30,31]. In this context, recent studies also referred to trust in authorities and government, anxiety, and the mental burden of parents and children and investigated the role of these factors in decisions about a potential COVID-19 vaccination uptake [28,32,33,34,35,36,37,38]. Because the population suffered mental distress during the pandemic, there is a need for more detailed and validated data on the extent to which mental burden impacts COVID-19 vaccination uptake [34]. In addition, there is still relatively little data on the impact of COVID-19 vaccination campaigns on vaccination willingness in recent years.

This work aims to consider these potentials and provide information on the impact of COVID-19 vaccination campaigns and political perceptions during the pandemic and to offer a perspective for improving future vaccination invitations concerning childhood vaccination backlogs or emerging pathogens. The objective of this paper is to analyze determinants and beliefs associated with the COVID-19 vaccination willingness for children among parents in Germany. Belief selections are related to the COVID-19 vaccine for children whereas determinants include the amount of mental burden, general attitudes towards vaccinations, and SARS-CoV-2 politics perceptions.

## 2. Materials and Methods

### 2.1. Study Design and Participants

This cross-sectional study was based on an online survey designed with the software Unipark EFS 21.2 (Tivian XI GmbH, Cologne, Germany). The investigation period was from 14 December 2021 to 27 January 2022. Survey distribution occurred via social media, notice boards, teachers, and chats in primary schools, general practitioners, pediatricians, and pharmacies in Germany. Informed consent was obtained digitally. Participation was voluntary, anonymous, and could be withdrawn at any time. Data were used only from completed surveys. Inclusion criteria were (1) being a parent, (2) being ≥18 years of age, and (3) having children between the ages of 5 and 11 years. After application of the selection criteria, 2401 (96.77%) of the 2481 complete data sets were eligible for further analysis. Overall, the study had a completion rate of 75.75% and a total number of 3275 participants. The time required to fill in the survey was approximately 13 min. The study was approved by the local Ethics Committees of the Faculty of Medicine of the University Hospital Essen (20-9307-BO).

### 2.2. Measures and Instruments

Data were collected on sociodemographic and mental health status, general vaccination attitudes, and SARS-CoV-2 politics perceptions during the COVID-19 pandemic and on beliefs leading to parents’ vaccination decisions. The following psychometric instruments were used to obtain information about parents’ health status: the Distress Thermometer (DT) was used to define the degree of experienced psychological distress in the past week (0 = no distress; 10 = extreme distress) [39,40]. A score of ≥4 indicates elevated distress [34,39]. The General Anxiety Disorder Scale-7 (GAD-7) assessed the frequency of general anxiety symptoms and the Patient Health Questionnaire (PHQ-8) captured depression symptoms over the past two weeks each on a 4-point Likert-scale (0 = never to 3 = nearly every day) [41]. GAD sum scores of ≥ 5, ≥10, and ≥15 indicate mild, moderate, and severe general anxiety symptoms [42,43,44]. PHQ sum scores of ≥5, ≥10, ≥15, and ≥20 indicate mild, moderate, intermediate, and severe depression symptomatology [41].

Accordingly, to capture parents’ general vaccination attitudes, the following self-generated items were implemented. These included completeness of vaccination certificates (for parents and children), the importance of vaccinations as protection against other possible infections (0 = strongly disagree; 100 = completely agree), and the present COVID-19 immunization status of parents.

Parents rated the presence of COVID-19-related fear, on a 7-point Likert-scale (1 = very low to 7 = extremely high) [34,40,45]. A score ≥ 5 relates to elevated COVID-19 fear [34]. Parents’ subjective level of information regarding COVID-19 covers “I feel informed about COVID-19”, “I feel informed about measures to avoid an infection with COVID-19” and “I understand the health authorities’ advice regarding COVID-19”. Items were evaluated on a 7-point Likert scale (1 = strongly disagree; 7 = completely agree). A score ≥ 5 relates to an elevated subjective level of information regarding COVID-19 [34]. Trust in governmental actions to face COVID-19 comprises “I think Germany is well prepared to face COVID-19”, “I think all government measures are being taken to combat COVID-19”, “I have confidence in the governmental system in Germany”, and “I believe that political actions against COVID-19 in Germany are exaggerated”. These variables were assessed on a 7-point Likert-scale (1 = strongly disagree; 7 = completely agree) [34,40,45]. The cut-off for elevated trust in governmental actions to face COVID-19 is set at ≥5 [34]. The questionnaires comprising COVID-19-related fear, subjective level of information regarding COVID-19, and trust in governmental actions facing COVID-19 are validated instruments and have been frequently used in previous surveys investigating COVID-19-related issues [34,35,38,40]. As an indicator of internal consistency, the reliabilities for the subjective level of information and governmental trust were evaluated using Cronbach’s α. Both scales showed an acceptable to high internal consistency with Cronbach’s *α* = 0.811 and Cronbach’s *α* = 0.676, respectively. Their scale–scale correlation was *r* = 0.497, *p* < 0.001.

The following self-generated items also contributed to the assessment of parents’ SARS-CoV-2 politics perceptions during the COVID-19 pandemic. Parents rated their pressure perception by COVID-19 vaccination campaigns on a numeric scale (0 = not at all; 100 = very strongly). They evaluated their experienced restrictions due to COVID-19 protective measures in their social environment (very little to very strong). The frequency (never to regularly) with which parents were politically engaged (e.g., at public demonstrations or in online forums and social media) against COVID-19-specific protective measures or legislative changes during the pandemic in the past seven months was also queried. Furthermore, parents’ attitude toward the adopted German measles protection law (mandatory vaccination since March, 1st 2020 addressing for example children attending public day care facilities) was assessed (fully adequate to completely overdone) [46].

Apart from that, probable beliefs contributing to COVID-19 vaccination decisions for children were created and included in the survey. Beliefs were subdivided into three multiple-response categories. Subsequently, filtered beliefs were either favoring vaccination, voting against vaccination, or beliefs that fit to the already received vaccination category. Parents selected individually appropriate beliefs depending on their vaccination decision. See the following Table 1 for further information regarding parents’ selectable beliefs.

### 2.3. Statistical Analysis

Data analysis was performed by using SPSS Statistics 28 software (IBM, Armonk, NY, USA). Descriptive statistics were conducted first. Sum scores for GAD-7 and PHQ-8 as well as mean scores for DT, importance of immunizations as protection against other possible infections, COVID-19-related fear, subjective level of information, governmental trust, and perceived pressure due to COVID-19 vaccination campaigns were calculated. Internal consistencies between subjective level of information and governmental trust were determined using Cronbach’s α. Correlations were performed to analyze associations between each selected variable and the willingness to vaccinate children against COVID-19. The strengths of associations were determined by Cramer-V, Cramer-Phi, Eta, Kendall’s Tau, and Spearman’s Rho. The correlation analysis findings are classified into different association ranges as per the selected coefficient. Association ranges differ as follows. Cramer’s V is distributed into weak < 0.2, moderate 0.2–0.6, and strong > 0.6 associations. The weak range of Cramer’s Phi is 0.1–0.3, the moderate range is 0.3–0.5 and the strong range is >0.5. Eta correlation results are classified as weak < 0.1, moderate 0.1–0.3, strong 0.3–0.5, and very strong > 0.5. The level of significance was set at *α* = 0.05 (two-sided test). To avoid the accumulation of alpha errors, *p* values were Bonferroni corrected after testing. Considering the present sample size, normal distribution was assumed.

## 3. Results

### 3.1. Parents’ Descriptive Characteristics Regarding Mental Health, General Vaccination Attitudes and SARS-CoV-2 Politics Perceptions

The sample included a total of 2401 complete data sets. Most of the participants were female (93.8%, *n* = 2252) and married (83.6%, *n* = 2007), the mean age was 38.95 years (range 20–59 years, SD = 5.010) and 57.2% of parents had a university degree (*n* = 1367). Overall, 64.6% (*n* = 1550) of parents had one child, most often being 5 years old (24.2%, *n* = 766). See Table 2 and Appendix A for parents’ sociodemographic data.

Overall, the majority of parents in this study indicated a positive general attitude towards vaccinations and COVID-19 vaccinations (see Table 3). A total of 71.4% (*n* = 1714) of parents preferred the COVID-19 vaccine for their children, while 19.4% (*n* = 465) of parents were reluctant to vaccinate their children. Table 3 provides details about parents’ general vaccination attitudes and SARS-CoV-2 politics perceptions.

### 3.2. Beliefs Contributing to COVID-19 Vaccination Uptake or Hesitancy

Parents approving vaccination (with and without STIKO recommendation and undecided, *n* = 1069) predominantly intended to protect children from infection in everyday life (for example kindergarten, school) (87.1%, *n* = 915) and feared long COVID-19 (83.3%, *n* = 875) (see Appendix A). Parents who denied COVID-19 vaccination uptake (*n* = 465) expected a less severe COVID-19 infection course in children (82.4%, *n* = 383). Also, they were not aware of the long-term effects (81.9%, *n* = 381). The most important beliefs for parents to approve COVID-19 vaccine uptake for their children prior to official recommendation (*n* = 863) were to achieve sufficient vaccination protection at an early stage (89.3%, *n* = 770), to reduce infections in daycare/school and to avoid quarantine (88.7%, *n* = 765). For details regarding beliefs contributing to vaccination decisions, see Appendix A.

### 3.3. Determinants Associated with COVID-19 Vaccination Willingness for Children

To determine the association of certain variables with COVID-19 vaccination willingness for children, several correlations were conducted. Table 4 highlights the variables exhibiting positive significant correlations with vaccination willingness and their adjusted *p*-values. Apart from gender (*r* = 0.055; *p*_adj_ > 0.999), residence area (*r* = 0.044; *p*_adj_ > 0.999), DT (*r* = 0.038; *p*_adj_ > 0.999), and PHQ-8 (*r* = 0.057; *p*_adj_ > 0.9), all variables demonstrated significant associations with vaccination willingness (*p* < 0.05). Additionally, there was an association of mandatory measles vaccination with perceived pressure by COVID-19 vaccination campaigns (Spearman *r* = 0.551; Kendall’s Tau-c = 0.371; *p* < 0.001).

When correlating vaccination willingness with determinants, cross-tabulations were created between parents with vaccination preference and vaccination reluctance (see Appendix A for distribution). Vaccine-accepting parents were aged ≥ 39 years (58%, *n* = 989), were married (85.2%, *n* = 1461) and held a university degree (61.8%, *n* = 1056 (see Appendix A). Vaccine-refusing parents were mainly aged < 39 years (60.6%, *n* = 280), were mostly married (76.1%, *n* = 354) and held a university degree in 44.2% (*n* = 203). General anxiety symptoms were predominantly absent at scores < 5 in 52.2% (*n* = 895) of those willing to vaccinate and in 45.2% (*n* = 210) of those refusing vaccination (see Appendix A).

Among vaccine-accepting parents, 90.9% (*n* = 1558) had complete vaccination certificates and 94.5% (*n* = 1619) had complete vaccination certificates for their children (see Appendix A). A total of 99.6% (*n* = 1707) of vaccine-accepting parents considered immunizations to be important for protection against other infections (scores ≥ 50). 99.9% (*n* = 1713) of parents preferring vaccination for their children had received COVID-19 immunizations prior to the present survey.

Vaccination acceptance was observed to be increased (scores ≥ 5) among individuals who reported elevated COVID-19-related fear (93.6%, *n* = 1605). Increased (scores ≥ 5) subjective COVID-19 knowledge was evident among both vaccine-accepting (97.0%, *n* = 1663) and vaccine-refusing parents (63.4%, *n* = 295) (see Appendix A). Lower (scores < 5) governmental trust was evident among both vaccine-accepting (69.8%, *n* = 1197) and vaccine-refusing parents (95.7%, *n* = 445). Vaccine-refusing parents mainly showed an increased (scores ≥ 50) perceived pressure by COVID-19 vaccination campaigns (87.7%, *n* = 408), heightened experience of restrictions in their social environment (83.6%, *n* = 389), and engagement against COVID-19 protective measures (51.6%, *n* = 240) and a negative attitude towards the mandatory measles law (57%, *n* = 265) (see Appendix A). Parents perceiving lower pressure (scores < 50) and parents perceiving higher pressure (scores ≥ 50) from COVID-19 vaccination campaigns both predominantly indicated the measles protection law was fully adequate to adequate (94.9% and 43.3%) (see Appendix A).

## 4. Discussion

This study analyzed beliefs and determinants associated with parents’ COVID-19 vaccination decisions for their underage children. Moreover, the goal of the study was to draw analogies and suggest improvements for future vaccination acceptance among parents. The descriptive results revealed a mild mental burden, a positive vaccination attitude in general, and a variation in perceptions of SARS-CoV-2 politics. Compared to the low national COVID-19 vaccination rate of 5–11-year-olds in Germany, the current findings revealed a high willingness to vaccinate children against COVID-19 (71.4%, *n* = 1714). This disparity may be explained by the STIKO’s revised vaccination advice, restricting the COVID-19 vaccination to minors with underlying medical conditions [13,14]. In addition to other determinants, it can be inferred that parents adhere to these updated vaccination recommendations.

Except for gender, residence area, DT, and PHQ-8, all determinants showed significant associations with vaccination willingness in this study. In line with previous research, significant correlations were exhibited between vaccination willingness and educational level, marital status, and age [25,26,36,47,48]. The correlation between age and COVID-19 vaccination willingness was found to be moderate (*r* = 0.208; *p*_adj_ < 0.05). However, the data suggested that older parents were more inclined to vaccinate their children compared to younger parents. Contrary to previous investigations, there was no significant correlation between vaccination willingness and gender [49,50,51,52]. Another distinction lies in the overall elevated level of education among vaccine-refusing and vaccine-accepting parents of this survey [48]. These findings are consistent with current research conducted in Germany, implying that COVID-19 vaccination uptake is associated with higher levels of education, marriage, and older parental age [53,54].

In terms of health status, the results indicated a mild mental health burden among participants. Although past investigations have demonstrated an association between mental health parameters and COVID-19 vaccination willingness during the pandemic, this study identified only a weak but significant association with parental general anxiety (*r* = 0.080; *p*_adj_ < 0.05) [48,55,56]. However, there was a significant correlation between vaccination willingness and COVID-19-related fear. The fear of COVID-19 was particularly elevated among vaccine-accepting parents, which has also been described in other studies [36,37]. This aligns with the provided vaccination beliefs, as vaccine-accepting parents mainly feared long COVID-19 and aimed for protection in everyday life. Moreover, a COVID-19-related fear has previously been declared as non-pathological in other research and could be explained by the novelty of the virus and the unknown aspects of COVID-19 [34,37].

In the category that addresses general vaccination attitudes, completeness of vaccination certificates and status of parents’ COVID-19 immunizations had moderate (*r* = 0.364; *p*_adj_ < 0.05 and *r* = 0.291; *p*_adj_ < 0.05) and very strong (*r* = 0.768; *p*_adj_ < 0.05) associations with vaccination willingness, respectively. Previous meta-studies from Greece and China considered the completeness of (children) vaccination certificates to be important predictors of the vaccination willingness (OR: 2.1517, 95% CI: 1.2181, 2.7696) [25,26]. Furthermore, previous results indicated that parents who were COVID-19 immunized were also willing for their children to receive the vaccine. This finding is consistent with other results in the literature [25,26,36,57,58]. Fittingly, there was a very strong positive correlation between parents’ perceived importance of vaccination as protection against other infections and the COVID-19 vaccination willingness (*r* = 0.734; *p*_adj_ < 0.05). The overall positive vaccination attitude of parents in the study aligns with previous research [25]. In summary, COVID-19 vaccination decisions are assumed to follow the same familial patterns as other vaccination decisions.

Regarding SARS-CoV-2 politics perceptions, determinants showed moderate and strong associations with vaccination willingness (see Table 4). Prior investigations have demonstrated associations between governmental trust and vaccine hesitancy and between the subjective level of COVID-19 knowledge and vaccine uptake during the pandemic [36,47,48,53,59,60,61]. This study came to a different conclusion because participants overall had high subjective levels of COVID-19 knowledge and low governmental trust. However, there exists a substantial amount of misinformation about COVID-19, the vaccines, and their effectiveness [25,30,31,53,61]. Given the uncertainty about the severity of the COVID-19 courses, premature conclusions by authorities on vaccination recommendations for a vulnerable group such as children should be discouraged. As trust is one of the major drivers for vaccine acceptance, valid and evidence-based information should be published and the government should aim to improve trust and reduce vaccine hesitancy [31,36,53,61,62,63,64].

This study further found that predominantly COVID-19 vaccine-refusing parents publicly opposed COVID-19-related protective measures and experienced restrictions in their social environment during the pandemic. However, physical restrictions could be avoided by the COVID-19 vaccines, as vaccinated families would have more freedom in their daily lives and this dimension of restriction they experienced would be reduced.

Another determinant very strongly associated with vaccination willingness was the pressure perception by COVID-19 vaccination campaigns (*r* = 0.775; *p*_adj_ < 0.05). Recent investigations examining vaccination willingness recommend providing additional detailed information and education about COVID-19 vaccination to increase knowledge and thus vaccination acceptance in the population, often through vaccination campaigns [26,36,53,63,65]. However, earlier findings of this study suggested elevated vaccination refusal among individuals who perceived increased pressure from vaccination campaigns and calls for COVID-19 vaccines. This aligns with a research study conducted in Poland, which reveals contrary outcomes when parents perceive excessive force through campaigns [66]. Therefore, it is not recommended to promote vaccine uptake solely through government-run vaccination campaigns, but also by extending education and developing sub-campaigns through other channels, not at the “helicopter” level, but at the family level, for example, pediatricians, general practitioners, health insurance companies, and schools [57,63]. However, for vaccination campaigns to have the desired effect, trust in public health policy and authorities may be a fundament. In outpatient settings, such as family medicine or general practice, motivational interviewing or other communication tools could be used when addressing vaccination.

The variable comprising pressure perception from COVID-19 vaccination campaigns additionally correlated with the attitude towards German mandatory measles vaccination. A vaccination campaign is an invitation to protect oneself against a possible pathogen, whereas a mandatory vaccine is prescribed by law [67]. However, a substantial proportion of parents who indicated increased pressure from COVID-19 vaccination campaigns predominantly had positive attitudes towards the mandatory measles vaccine. This result could be explained by the fact that parents consider measles and measles vaccinations to be established, whereas vaccination against COVID-19 is a novelty [48,68].

Beliefs favoring vaccination for children were related to protecting children during everyday life, to fear of COVID-19 and long COVID-19. The main beliefs of those parents denying COVID-19 vaccines for their children were the expectation of a less severe course of infection among children and the missing of information about the long-term effects of COVID-19 vaccines. These beliefs are generally following the existing literature, although there may be nuances in the ranking [25].

Consistent with previous research, various factors are associated with the COVID-19 vaccination uptake decision [36,48]. Numerous investigations have analyzed parental COVID-19 vaccination willingness for their children through sociodemographic data, general attitudes towards vaccinations, trust in the government, and vaccines. This study contributes to the existing research on COVID-19 vaccination willingness by examining beliefs and factors related to mental health, COVID-19 fear, compulsory measles vaccination, and perceptions of SARS-CoV-2 politics. It considers the pandemic situation and incorporates parents’ perceptions of the political environment by measuring perceived pressure from COVID-19 vaccination campaigns, experienced restrictions in the social environment during the pandemic, and engagement against COVID-19 protective measures. Concerning low acceptance and numbers of childhood COVID-19 vaccinations, many studies recommend further education about vaccines, transparency, and the extension of campaigns. However, it should be considered that for many parents, this may increase perceived vaccination pressure, which can lead to an increased vaccination refusal. Vaccination campaigns are shown to have a major impact on target groups. Therefore, it is crucial to sensitively approach recipients when introducing new vaccines.

For this investigation, some limitations should be considered. The study only focuses on assumptions, but no causality is provided. There might be a selection bias because this survey was distributed via online and analog channels. A potential bias may apply to individuals with online access. Moreover, gender imbalance should be considered, and participants were predominantly of higher education. This cross-sectional study does not allow conclusions to be drawn about the general public and only assesses a snapshot of the COVID-19 pandemic. Other study designs could be applied to assess further details, improve vaccination pressure perception and trust, and generate data on societal pressures and community influence.

## 5. Conclusions

This study investigated the COVID-19 vaccination willingness among parents in Germany and analyzed which determinants as well as beliefs lead them to decide for or against COVID-19 vaccination uptake for children aged 5 to 11 years. Determinants comprised the mental health status of parents, the vaccination attitudes of parents, and their perception of German SARS-CoV-2 politics. Overall, this study demonstrates a high acceptance among parents to vaccinate their children against COVID-19. In summary, the results indicated that several factors are associated with vaccination willingness, but they differ in strength. Among parents accepting the COVID-19 vaccine for their children, the general vaccination attitudes were predominantly positive, and the majority feared COVID-19. Among the parents refusing the COVID-19 vaccine for their children, the perceptions of SARS-CoV-2 politics were mainly negative. Overall, data showed low governmental trust among parents. Mental health variables were the least associated with vaccination willingness in this study. It should be emphasized that parents who experienced pressure from COVID-19 vaccination campaigns were more reluctant to vaccinate their children against COVID-19 but had a positive attitude towards the mandatory measles vaccination. For the implementation of novel vaccination initiatives, it is crucial to acknowledge that parental attitudes may vary based on whether the vaccine targets a well-known or previously unknown pathogen. As health institutions warn of childhood vaccination backlogs, future vaccination acceptance strategies should address families, sensitively approach recipients, reduce pressure, and aim to improve trust. For additional measures to be built, trust in public health policies is essential. Sub-campaigns may target families or individuals, for example, in outpatient clinics or through social media. Further surveys should be initiated on vaccination certificates, vaccine importance, the prevalence of preventable childhood diseases, and political perceptions including trust. Future investigations on vaccination willingness should consider the potential impact of societal pressures or community influence on parental decisions.

## Figures and Tables

**Table 1 vaccines-12-00020-t001:** Selection of beliefs contributing to COVID-19 vaccination uptake and hesitancy for children.

Categories	Beliefs
**Parents who are willing to have their children** **vaccinated** ^1^	Protect child/children from infection in everyday life (daycare, school, etc.)
	Fear of a severe course
	Fear of “long COVID-19”
	Chronically ill/immunocompromised child/children
	Child/children want/desire vaccination
	Regain more freedoms
**Parents who already approved the vaccination of their** **children prior to the official** **recommendation**	Achieve sufficient vaccination protection at an early stage
	Reduce infections in daycare/school and avoid quarantine
	Regain more freedom
	Fear of a severe course
	Chronically ill/immunocompromised child/children
**Parents who refused to vaccinate their children**	I think that my child/children is/are too young
	I think that the vaccine is not well studied
	I am not aware of the long-term effects
	I think my child/children would pass an infection without severe progression
	My child/children has/have an allergy to vaccine components, thrombosis-with-thrombocytopenia syndrome, or capillary leak syndrome
	Corona pandemic is an adult pandemic, which affects children less
	COVID-19 infection is less severe in children
	My child/children do not belong to particularly vulnerable or pre-diseased groups
	My child/children refuse/refuse vaccination

^1^ Parents who are willing to have their children vaccinated with and without the Standing Committee on Vaccination recommendation and who are undecided. Their children are not (yet) vaccinated.

**Table 2 vaccines-12-00020-t002:** Sociodemographic characteristics of parents (*n* = 2401).

	*n*	%
**Age**		
18–24	5	0.2
25–34	433	18.0
35–44	1648	68.6
45–54	300	12.5
55–64	4	0.2
**Gender**		
Female	2252	93.8
Male	149	6.2
**Marital Status**		
Single	73	3.0
Married	2007	83.6
In a relationship	223	9.3
Separated/divorced	93	3.9
Widowed	3	0.1
Other	2	0.1
**Level of education**		
University degree	1367	57.2
High school degree	315	13.2
Higher middle school degree	105	4.4
Lower middle school degree	597	25.0
Other forms of schooling	4	0.2
**Residence area**		
100,000 residents	852	35.5
20,000 residents	629	26.2
5000 residents	426	17.7
<5000 residents	494	20.6
**Number of children**		
1	1550	64.6
2	734	30.6
3	96	4.0
More than 3	21	0.9
**Age of children**		
5	766	24.2
6	539	17.1
7	478	15.1
8	413	13.1
9	358	11.3
10	301	9.5
11	305	9.7
**Health status**		
Diseases	695	28.9
Physical disease	486	20.2
Mental disease	209	8.7
No disease	1706	71.1
**Mental health status**	**M**	**SD**
DT	5.7	2.654
GAD-7	5.3	4.296
PHQ-8	5.6	4.338

**Table 3 vaccines-12-00020-t003:** Parents’ general vaccination attitudes and SARS-CoV-2 politics perceptions.

General Vaccination Attitudes	*n*	%
**Vaccination certificate (parents)**		
Complete	1891	78.8
Partial complete	309	12.9
Incomplete	195	8.1
So far, the COVID-19 vaccination is the only vaccination I have ever received.	6	0.2
**Vaccination certificate (children)**		
Complete	2048	85.3
I do not know	9	0.4
Incomplete	288	12
My child has not received any childhood vaccinations.	56	2.3
**Importance of immunizations as protection against other infections**	84.89 ^1^	26.464 ^2^
**COVID-19 vaccination of parents**		
Yes	2084	86.8
No	317	13.2
**Willingness for COVID-19 vaccination of children aged 5 to 11 years**		
Yes, if a Standing Committee on Vaccination recommendation is available	430	17.9
Yes, even if no Standing Committee on Vaccination recommendation is available	1284	53.5
Undecided	222	9.2
No, in no case	465	19.4
**Children (5 to 11 years old) who already received the vaccination** **COVID-19 vaccination**		
Yes	863	44.7
No	1069	55.3
**SARS-CoV-2 politics perceptions**	**M**	**SD**
COVID-19-related fear	5.3	1.683
Subjective level of information	6.0	1.082
Governmental trust	3.9	1.199
Pressure perception by COVID-19 vaccination campaigns	26.5	37.741
**Experienced restrictions in the social environment**	**n**	**%**
Very little	290	12.1
Little	881	36.7
Neither	382	15.9
Strongly	579	24.1
Very strong	269	11.2
**Engagement against protective measures**		
Never, and do not plan to	1905	79.3
Never, but plan to do so	105	4.4
Once	57	2.4
Several times	228	9.5
Regularly	106	4.4
**Attitude towards mandatory measles vaccine**		
Fully adequate	1516	63.1
Adequate	433	18.0
Neutral	158	6.6
Exaggerated	154	6.4
Completely overdone	140	5.8

^1^ Mean score, ^2^ Standard deviation.

**Table 4 vaccines-12-00020-t004:** Results of correlations between selected variables and the COVID-19 vaccination willingness for children.

	Cramer V ^2^	Cramer Phi ^3^	Eta ^4^	CI (95%)	Unadjusted *p* Values	Adjusted *p* Values ^1^
**Sociodemographic**						
Age			0.208	(0.171–0.251)	<0.001	0.019
Gender	0.055			(0.026–0.1)	0.063	>0.999
Marital Status	0.071			(0.057–0.105)	<0.001	0.019
Educational level		0.198		(0.173–0.252)	<0.001	0.019
Residence Area			0.044	(0.004–0.083)	0.203	>0.999
**Mental health status**						
DT			0.038	(0.015–0.083)	0.328	>0.999
GAD-7			0.080	(0.044–0.123)	0.002	0.038
PHQ-8			0.057	(0.029–0.1)	0.05	0.95
**General vaccination** **attitudes**						
Vaccination certificate (parents)	0.364			(0.343–0.386)	<0.001	0.019
Vaccination certificate (children)	0.291			(0.265–0.318)	<0.001	0.019
Importance of immunizations as protection againstother infections			0.734	(0.707–0.760)	<0.001	0.019
COVID-19 vaccination of parents	0.768			(0.734–0.8)	<0.001	0.019
**SARS-CoV-2 politics** **perceptions**						
COVID-19-related fear			0.665	(0.635–0.698)	<0.001	0.019
Subjective level of information			0.484	(0.445–0.522)	<0.001	0.019
Governmental trust			0.602	(0.572–0.631)	<0.001	0.019
Pressure perception by COVID-19 vaccination campaigns			0.775	(0.748–0.799)	<0.001	0.019
Experienced restrictions in the social environment		0.642		(0.606–0.680)	<0.001	0.019
Engagement against protective measures		0.614		(0.581–0.654)	<0.001	0.019
Attitude towards mandatory measles vaccine		0.785		(0.754–0.815)	<0.001	0.019

^1^ Bonferroni correction was applied for 19 tests at α = 0.05. ^2^ Cramer V: weak: <0.2, moderate: 0.2–0.6, strong: >0.6. ^3^ Cramer Phi: weak: 0.1–0.3, moderate: 0.3–0.5, strong: >0.5. ^4^ Eta: weak: <0.1, moderate: 0.1–0.3, strong: 0.3–0.5, very strong: >0.5.

## Data Availability

Data are available on request due to privacy restrictions.

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
