# Peer review of "COVID-19 Vaccine for Children: Determinants and Beliefs Contributing to Vaccination Decision of Parents in Germany 2021/2022"

_vaccines, 2023, doi:10.3390/vaccines12010020_

Round 1

Reviewer 1 Report

Comments and Suggestions for Authors

The authors have conducted a cross-sectional study to determine and   factors that weighed on  parents' willingness to vaccinate their children  aged 5-11 years against COVID-19 in Germany. Among the the determinants  investigated were the parents' mental health status, vaccination attitudes, and perception of German SARS-CoV-2 politics. Online surveys were conducted using a structured questionnaire, as well as direct interviews after informed consent. The mental health status was surveyed using standard methods. The unique merit of the present study was the quantitative determination of the parents' mental health status as driver of vaccine hesitancy. Statistical analysis of the results showed a high vaccine acceptance rate. the major factor driving vaccine hesitancy and rejection was fear of the safety of the COVID-19 vaccine and undue pressure from Government politics.

Overall, this is a well designed and well executed study with interesting insights especially about the mental health status of targeted children's  parents. There were some grammatical slips eg line 16  ,write "to reduce the number of cases..."instead of "to reduce the number of "courses...", which can be corrected on careful proofreading of the final text.

With this correction I recommend the article for publication in MDPI Vaccines.

Comments on the Quality of English Language

Line 16: there seems to be some confusion between the use of the words cases and courses interchangeably. The authors should verify and correct as necessary. Otherwise  the language is fine.

Reviewer 2 Report

Comments and Suggestions for Authors

Purrmann et al. reported a survey of 2401 parents in Germany on the determinants and beliefs that relate to their vaccination decision for their children of 5~11 years old. Overall, there is a high advocation of COVID vaccine among parents for the children. Major variables on supporting the vaccination include parents’ vaccination certificate, covid-19 immunizations, and increased covid fear. Major variables on refusing the vaccination include pressure from vaccine campaigns, restrictions due to covid protective measures, and engagement against covid protective measures. Although this is an internet survey which may have bias to people with online access, it still provide valuable information for the interest of general public as well as the government on the establishment of future health-related policy which is not restricted to COVID.

1.     Line 72~73: Was this ‘at least one dose’ or ‘only one dose’?

2.     For Table 4: Is there a positive or negative correlation between the variables and the willingness for vaccination? If so, I recommend the authors to label them in the table unless they are all positive or all negative correlations. Can authors elaborate it by describing some examples, such as educational level, marital status and age as mentioned in Line 279.

3.     Line 297~298: Can authors define what is ‘medium’ and ‘strong’ association? Maybe define it in the method as well as in the table. Put the numbers here in the sentences to remind readers. In addition, replace ‘to’ by ‘and..., respectively’

4.     Line 308~309: similar to the previous point.

5.     Can authors elaborate in the discussion about how representative are participants in the survey compared to the country? For example, as the authors mentioned, majority of the participants have college degree. What is the percentage of females with college degree in Germany? Are there other studies in Germany on COVID advocacy that authors can use to compare to the current study?

6.     Did authors possibly consider some other variables that may affect vaccine opinions, such as locations (city vs countryside), ethnicity, family income…?

Reviewer 3 Report

Comments and Suggestions for Authors

The authors have made an interesting attempt at “COVID-19 Vaccine for Children: Determinants and Beliefs Contributing to Vaccination Decision of Parents in Germany 2021/2022”. The manuscript is interesting; however, the authors need to justify the scientific writing manuscript. Some of the general comments are provided below:

1.      What are the potential biases introduced by using an online survey? Are there specific demographics or viewpoints that might be overrepresented or underrepresented in this sampling method?

2.     Considering the high percentage of female participants (93.8%) in the study, how might this gender imbalance affect the generalizability of the findings? Could gender-specific attitudes towards vaccination have an impact on the results?

3.     What factors might contribute to the discrepancy between parents who prefer COVID-19 vaccination for their children (71.4%) and those who are hesitant (19.4%)? Are there specific concerns or beliefs that predominantly influence each group's decision?

4.     How do the beliefs identified in the study align with public discourse or other research findings regarding COVID-19 vaccination for children? Are there additional beliefs or factors that might have been overlooked in this analysis?

5.     How might societal pressures or community influence impact parents' vaccination decisions, especially concerning perceived pressure from vaccination campaigns and attitudes toward mandatory vaccination laws?

6.     The study highlights lower governmental trust among vaccine-refusing parents. What implications might this have for public health policies and communication strategies related to COVID-19 vaccination campaigns?

7.     How do the findings of this study align with or differ from previous research on parental attitudes towards childhood vaccinations, especially concerning COVID-19? What might explain discrepancies or similarities between this and other studies?

8.     What implications do these findings have for public health policies aimed at increasing childhood vaccination rates, especially in the context of COVID-19? How might policies address the nuances observed in parental attitudes and beliefs?

Reviewer 4 Report

Comments and Suggestions for Authors

The manuscript “COVID-19 Vaccine for Children: Determinants and Beliefs Contributing to Vaccination Decision of Parents in Germany 2021/2022” deals with a quite interesting topic, which is the willingness of parents to vaccinate children aged 5-11 against COVID-19. Although the topic might not be exactly innovative, the manuscript is adequately presented and structured. Below are some considerations:

Introduction

I would recommend expressing the objectives of the study more clearly, also with a view to any future perspectives. At present, there is more of a list of the determinants analyzed.

Materials and Methods

The authors should add information related to the validation of the questionnaire.

Discussion

I would suggest the authors to further develop the discussion of the results, in particular by comparing the results obtained with the vaccination rates reported at a national level.

Conclusions

Conclusions should be more concise and focus mainly on the main results obtained and on the future perspectives.

Comments on the Quality of English Language

Minor editing of English language required.

Reviewer 5 Report

Comments and Suggestions for Authors

This is a very good manuscript and I encourage publication. The methodology is sound and well explained, should others want to replicate this study in the future, and the presentation of the results is very clear. The bibliography is really up to date and exhaustive and the manuscript is very well written and structured.

The sample of parents in this study in evidently biased as it is not a random sample of the parent population of Germany but interested participants were enrolled as participants by a general snowballing mechanism by various means. However, that caveat does not cause problems for the overall validity of the study as the authors do not make any claims about the the views of the larger population of Germany.

One of the findings of the study that really struck me was a general low governmental trust expressed by respondents overall, while the was a clear difference in the levels of trust in COVID-19 policies between parents willing to have their children vaccinated and those who were hestitant. This lead me to what I think is the most important message of this manucripts. The authors conclude with some very sensible and actionable recommendations about how to design childhood vaccinations in the future by going beyond an exclusive reliance on guidelines issued by by governments (be the regional, national or more global) and incorporating the vaccines in the more local context of families, family doctors and nurses, nursery schools etc. 

Round 2

Reviewer 3 Report

Comments and Suggestions for Authors

The authors have addressed my queries and now the article is acceptable for publication. 

Reviewer 4 Report

Comments and Suggestions for Authors

I appreciate the effort made by the authors in answering my issues about the manuscript “COVID-19 Vaccine for Children: Determinants and Beliefs Contributing to Vaccination Decision of Parents in Germany 2021/2022”. However, I still have some concerns related to the materials and methods and the discussion section.

Materials and Methods

I thank the authors for clarifying the aspects related to the validation of the questionnaire. In any case, the line "based on past surveys" included in the text could be misleading. Please add a sentence that clearly indicates that the questionnaires have been validated in previous studies.

Discussion

The authors have correctly included a comparison with national data but a specific comment must also be included. Particularly as the study results differ widely from the national vaccination rates. The authors have correctly included among the limitations of the study the fact that the results are not generalizable, but it is necessary to include possible explanations for this large difference.

Comments on the Quality of English Language

Minor editing of English language required.
